# Acetylation of the yeast Hsp40 chaperone protein Ydj1 fine-tunes proteostasis and translational fidelity

Siddhi Omkar[1], Megan M. Mitchem[1], Joel R. Hoskins[2], Courtney Shrader[1], Jake T. Kline[3], Nitika[1], Luca Fornelli[3], Sue Wickner[2], Andrew W. Truman[1]*

**1** Department of Biological Sciences, The University of North Carolina at Charlotte, Charlotte, North Carolina, United States of America, **2** Laboratory of Molecular Biology, National Cancer Institute, National Institutes of Health, Bethesda, Maryland, United States of America, **3** School of Biological Sciences, University of Oklahoma, Norman, Oklahoma, United States of America

* a.truman@charlotte.edu

**Data Availability Statement:** The data can be found either in the manuscript and supplemental tables. The raw proteomic data can be found on the ProteomeXchange #PXD051428 (https://

## Abstract

Proteostasis, the maintenance of cellular protein balance, is essential for cell viability and is highly conserved across all organisms. Newly synthesized proteins, or "clients," undergo sequential processing by Hsp40, Hsp70, and Hsp90 chaperones to achieve proper folding and functionality. Despite extensive characterization of post-translational modifications (PTMs) on Hsp70 and Hsp90, the modifications on Hsp40 remain less understood. This study aims to elucidate the role of lysine acetylation on the yeast Hsp40, Ydj1. By mutating acetylation sites on Ydj1's J-domain to either abolish or mimic constitutive acetylation, we observed that preventing acetylation had no noticeable phenotypic impact, whereas acetyl-mimic mutants exhibited various defects indicative of impaired Ydj1 function. Proteomic analysis revealed several Ydj1 interactions affected by J-domain acetylation, notably with proteins involved in translation. Further investigation uncovered a novel role for Ydj1 acetylation in stabilizing ribosomal subunits and ensuring translational fidelity. Our data suggest that acetylation may facilitate the transfer of Ydj1 between Ssa1 and Hsp82. Collectively, this work highlights the critical role of Ydj1 acetylation in proteostasis and translational fidelity.

## Author summary

Cells require a suite of chaperone and co-chaperone proteins to maintain a healthy balance of functional proteins. A large number of modifications on chaperone and co-chaperone proteins have been identified, but their functional importance has not been fully explored. In this study, we identify acetylation sites on the yeast co-chaperone Ydj1 that impact its interactions with major chaperones and client proteins including those involved in protein synthesis. This work sheds light on how modifications on co-chaperones can also play an important role in the health of the proteome.

proteomecentral.proteomexchange.org/cgi/
GetDataset?ID=PXD051428).

**Funding:** This work was supported by the NIH
(R01GM149639 and R01GM139885 to AWT,
R35SGM147397 to LF), National Science
Foundation Graduate Research Fellowship
Program under Grant No. 2235055 (MMM), the
Intramural Research Program of the NIH, NCI,
Center for Cancer Research (SW). The funders had
no role in study design, data collection and
analysis, decision to publish, or preparation of the
manuscript.

**Competing interests:** The authors have declared
that no competing interests exist.

## Introduction

The highly dynamic and crowded cellular landscape presents a challenge for newly synthesized proteins that need to achieve and retain their active, native fold. Molecular chaperones such as Hsp70 and Hsp90 are directly involved in supporting protein function and assist in the assembly of multimeric complexes by binding "clients" as they are synthesized [1]. The activity and specificity of Hsp70 is directed by a suite of co-chaperone proteins primarily composed of Hsp40s, also known as J-domain proteins (JDPs) and nucleotide exchange factors (NEFs) [2,3]. In budding yeast, 22 JDPs provide support for 11 Hsp70 paralogs [4]. The most highly expressed of these is Ydj1 at over 40,000 molecules per cell [4]. Perturbation of Ydj1 function results in a variety of phenotypes that include poor growth at room temperature and severe temperature sensitivity [5,6]. Cells lacking Ydj1 are also sensitive to a range of cell wall damaging agents such as caffeine, calcofluor white (CFW), and sodium dodecyl sulfate (SDS), probably as a result of the interplay between chaperones and the yeast cell integrity MAPK pathway [7,8]. Ydj1 and its homologs contain six major functional domains that contribute to their crucial role in cellular protein folding processes [2,9,10] (Fig 1A and 1B). The N-terminally located domain (J-domain) is required for stimulation of the ATPase activity of Hsp70, facilitating the transfer of client proteins from Hsp40 to Hsp70 for further processing [11,12]. Studies of this region have identified a highly conserved sequence, His-Pro-Asp (HPD), essential for Hsp70-Hsp40 interaction [13,14]. This domain is followed by a Glycine/Phenylalanine-rich region, which helps determine client specificity [15]. Following the Glycine/Phenylalanine-rich region, there are two carboxy-terminal domains (CTDs), which act as key client-binding domains [16]. The first of these domains (CTDI) contains a client-binding site. Within the Ydj1 CTDI is a zinc-finger-like region, which aids in the stabilization of client proteins [17]. The second CTD, CTDII, also binds clients and is paramount for J-protein self-regulation [12,17]. This regulatory role is made possible with help from the dimerization domain. Following the dimerization domain of Ydj1 is a C-terminal extension. While its function is currently undefined, a similar region in human DNAJA2 regulates self-association and chaperone activity [18].

Despite several decades of research, the only post-translational modification on Ydj1 that has been extensively studied is C-terminal farnesylation [19–21]. While Ydj1 is primarily cytosolic, farnesylation at cysteine 406 facilitates the localization of Ydj1 to the Endoplasmic Reticulum and the perinuclear membrane [21,22]. This modification is crucial for the regulation and activation of specific Hsp90 clients [21]. Recent studies have shown that in mammalian cells, Hsc70 and its co-chaperone DNAJA1 are deacetylated by HDAC6, a process required for the interaction between the two proteins [23]. In this study, we have explored the impact of Ydj1 J-domain acetylation on its function both in vivo and in vitro. Our findings suggest that Ydj1 acetylation plays a vital role in proteostasis and translational fidelity.

## Results

### Ydj1 acetylation impacts the yeast stress response

Global proteomic experiments have identified 12 acetylation sites on Ydj1 (GPMdb, https://gpmdb.thegpm.org/). To prioritize sites for further study, we considered their conservation throughout nature and their location on important functional domains. Six acetylation sites, K23, K24, K32, K37, K46, and K48 are found on the J-domain of Ydj1, a region essential for interaction with Ssa1 and Ydj1 function [3,13,19] (Fig 1A and 1B). Interestingly, these sites are highly conserved in eukaryotes, suggesting potential functional importance [19] (Fig 1C). To assess the influence of acetylation on yeast stress resistance, we constructed a yeast centromeric

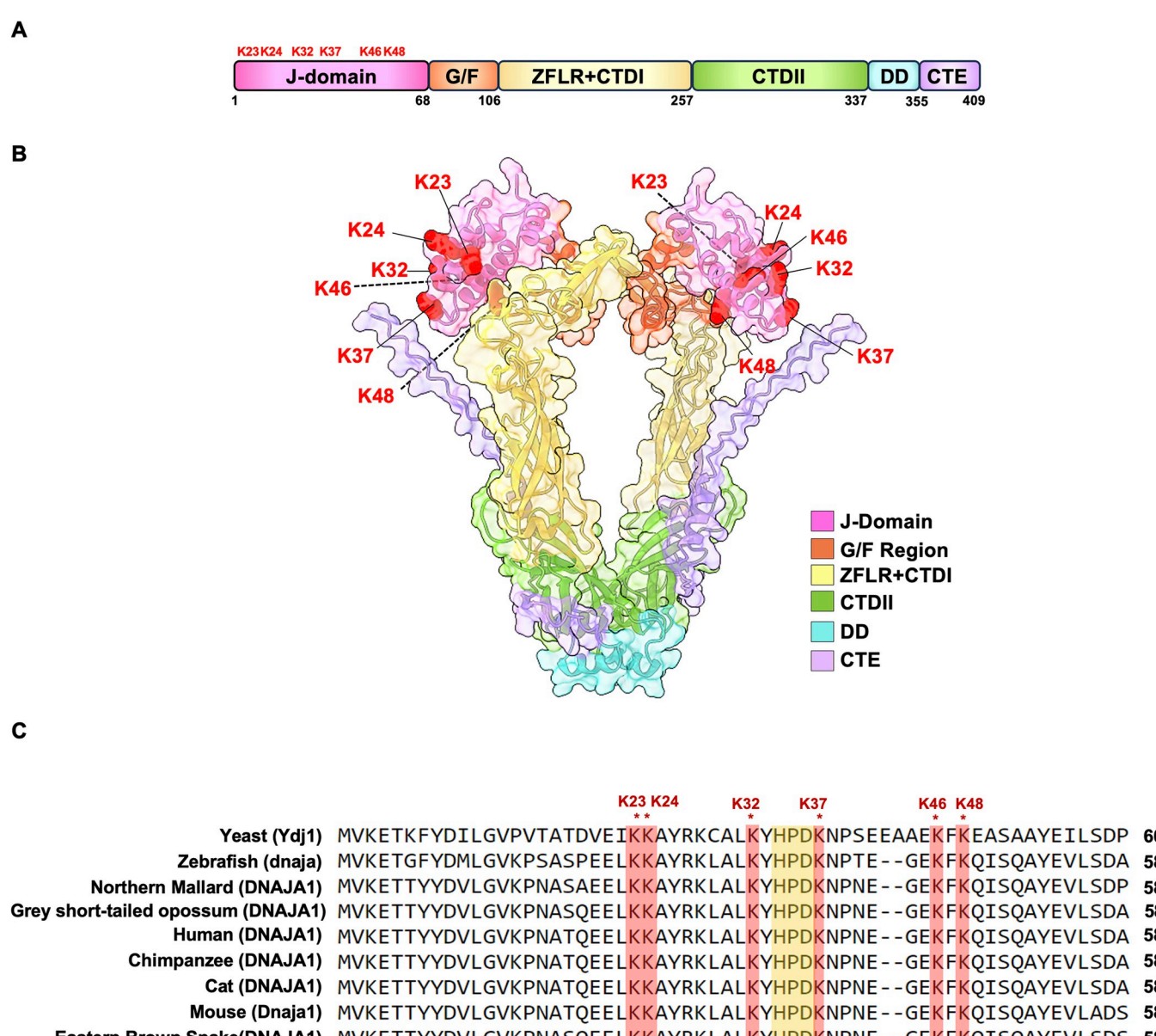

**Fig 1. Yeast Ydj1 contains six acetylation sites in its J-domain.** (A) Domain structure of yeast Ydj1. Sites of acetylation on the J-domain are highlighted in red. (B) Cartoon representation of Ydj1 with individual domains colored as indicated. Acetylated residues are labeled in red. This dimeric structure was generated by Alphafold3 and rendered in ChimeraX. (C) Alignment of Ydj1 homologs showing the location and conservation of J-domain acetylated lysines. The conserved HPD region important in Hsp70-J Protein interactions is highlighted in yellow.

plasmid expressing Ydj1 with a C-terminal FLAG tag from a constitutive GPD promoter. We individually substituted each of the six J-domain lysines with either arginine (non-acetylatable) or glutamine (acetyl-mimic) in a manner similar to previous studies [24–27]. Additionally, we introduced combined substitutions of all six J-domain lysines to either arginine or glutamine (hereafter denoted as 6KR or 6KQ, respectively). To investigate the impact of acetylation on Ydj1 function, *ydj1Δ* cells were transformed with plasmids carrying an empty vector control or expressing wild-type Ydj1 or acetylation site mutants (R and Q versions of K23, K24, K32, K37, K46, K48). Ydj1 mediates the response to a range of cellular stresses. As a co-chaperone

in the Hsp70-Hsp90 system Ydj1 is required for heat shock response [5,14,28]. Ydj1 also supports activity of the yeast Pkc1 pathway and thus resistance to cell wall perturbing agents that include caffeine, calcofluor white (CFW) and sodium dodecyl sulfate (SDS). Finally, the ribonucleotide reductase complex (RNR) is a client of Ydj1 in yeast [22,29]. Loss of Ydj1 results in destabilization of Rnr2 and Rnr4 and sensitivity to the RNR inhibitor hydroxyurea, HU [22]. Cells expressing wild-type and acetylation-site mutants were screened against heat shock, caffeine, calcofluor white (CFW), hydroxyurea (HU), and sodium dodecyl sulfate (SDS). While cells lacking Ydj1 displayed increased sensitivity to all stress conditions compared to the wild-type, those expressing any of the six arginine substitutions grew comparably to wild-type, indicating that preventing acetylation of residues in the Ydj1 J-domain minimally affects yeast stress resistance (Fig 2A). In contrast, substituting J-domain lysines with glutamine (KQ) had

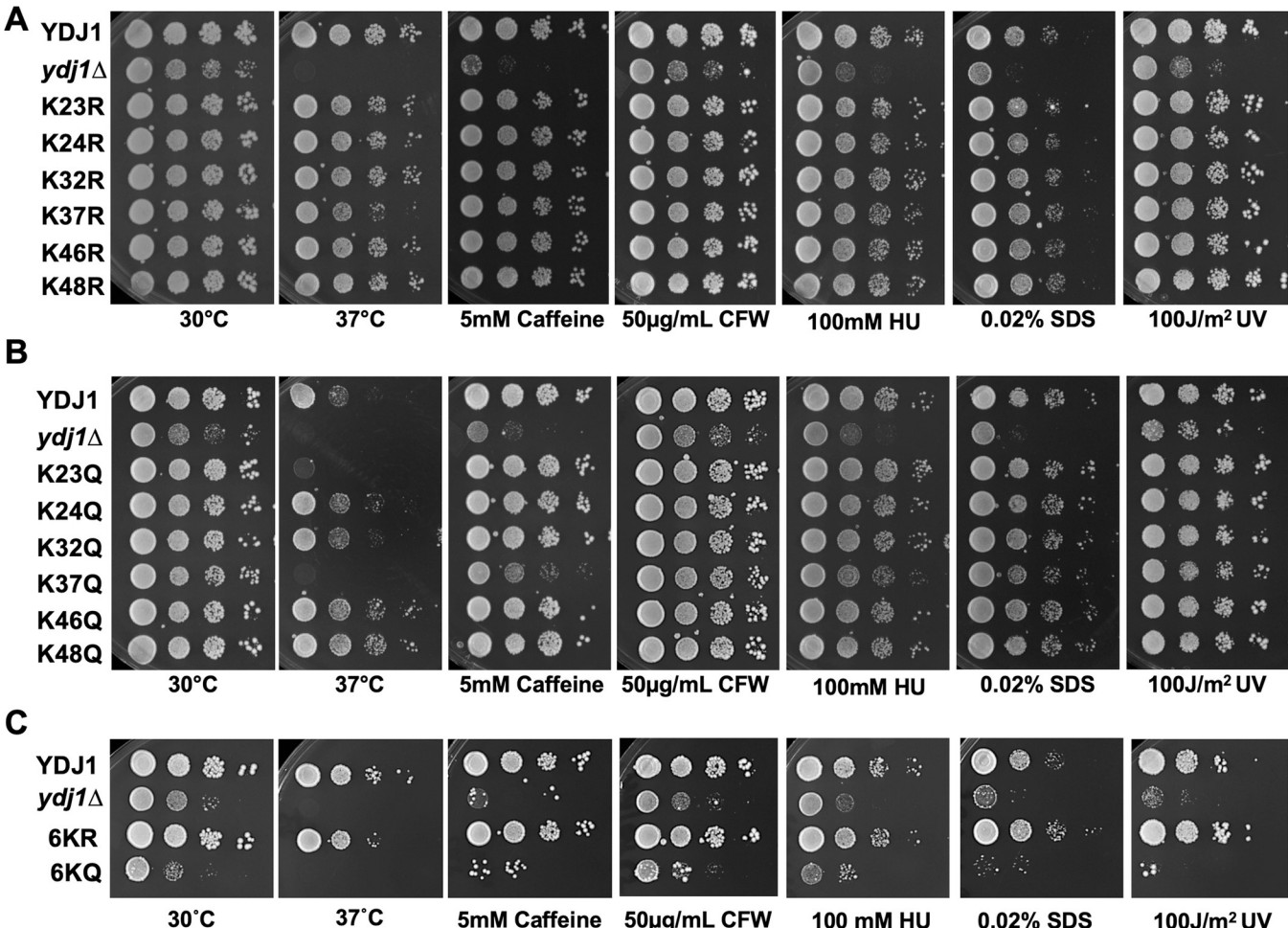

**Fig 2. Ydj1 acetylation impacts yeast stress resistance.** (A) Serial dilution of yeast expressing mutations that prevent acetylation of residues in the Ydj1 J-domain. *ydj1Δ* cells were transformed with either the empty control plasmid (pAG413) or plasmids expressing wild-type Ydj1 or Ydj1 single KR mutants. Transformants were grown to mid-log phase and then tenfold serially diluted onto YPD media containing the indicated stress agents. Plates were photographed after three days. (B) Serial dilution of yeast expressing mutations that mimic acetylation of residues in the Ydj1 J-domain. *ydj1Δ* cells were transformed with either the empty control plasmid (pAG413) or plasmids expressing wild-type Ydj1 or Ydj1 single KQ mutants. Transformants were grown to mid-log phase and then tenfold serially diluted onto YPD media containing the indicated stress agents. Plates were photographed after three days. (C) Serial dilution of yeast expressing 6KR/6KQ mutations. *ydj1Δ* cells were transformed with either the empty control plasmid (pAG413) or plasmids expressing wild-type Ydj1, 6KQ or 6KR mutants. Transformants were grown to mid-log phase and then tenfold serially diluted onto YPD media containing the indicated stress agents. Plates were photographed after three days.

a more pronounced effect on Ydj1 function. Notably, cells expressing Ydj1 K23Q and K37Q failed to thrive at elevated temperatures (Fig 2B). Interestingly, this effect was not universal across all tested stress conditions. Although K37Q displayed partial caffeine-sensitivity, it mirrored wild-type Ydj1 function under other stress conditions (Fig 2B). K23Q and K24Q cells grew at wild-type rates in response to caffeine, calcofluor white (CFW), SDS, and UV treatment (Fig 2B). Mutation of all six sites to arginine (6KR) had no discernible impact on the cellular response to the tested stresses (Fig 2C). Conversely, the 6KQ mutation produced the most notable loss-of-function phenotype, exhibiting sensitivity to all stresses examined (Fig 2C).

## Ydj1 acetylation alters Ydj1 abundance and mobility on SDS-PAGE

Acetylation can impact a range of protein properties, including protein stability [30]. Several of the Ydj1 KQ mutants were defective for growth under several conditions, suggesting that acetylation may alter Ydj1 stability. We analyzed the Ydj1 abundance in cells expressing Ydj1 wild-type and lysine substitutions from plasmids via Western blot analysis. The majority of the single KR mutants displayed levels of Ydj1 comparable to the wild-type except for K46R, which exhibited a subtle decrease in Ydj1 (Fig 3A). Similarly, the abundance of Ydj1 was not significantly impacted in the majority of single KQ mutants except for K37Q, which displayed a large increase in Ydj1 (Fig 3A, bottom gel). Interestingly, this was not observed in the 6KQ mutant, suggesting a potential cooperative effect between the six acetylation sites (Fig 3A, bottom gel). We also considered the possibility that exposure to stress conditions in Fig 2 may trigger degradation of Ydj1 mutants. To determine this, we assessed the steady state levels of the Ydj1 6KQ mutant in response to stress. 6KQ Ydj1 abundance was independent of all conditions tested, suggesting that the phenotypes observed in Fig 2 were not a result of protein degradation (Fig 3B). As seen previously, Ydj1 wild-type appeared as a doublet by Western blot analysis due to the presence of both unfarnesylated and farnesylated forms [31]. All of the Ydj1 mutants were also observed as doublet bands by Western blot analysis, indicating that

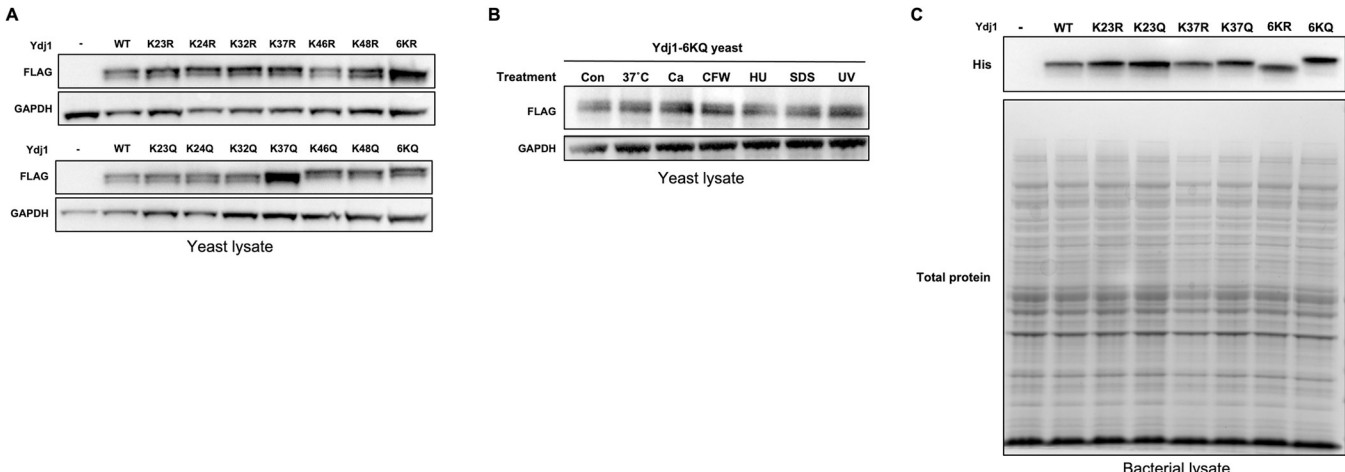

**Fig 3. Impact of acetylation on steady-state levels and gel mobility of Ydj1.** (A) Lysates from *ydj1Δ* yeast cells transformed with either a control plasmid or a plasmid expressing the indicated Ydj1-FLAG mutant grown to mid-log phase were analyzed by SDS-PAGE followed by Western Blotting using antisera to either FLAG or GAPDH (loading control). (B) Stress does not trigger degradation of Ydj1 6KQ. 6KQ yeast cells were treated with the indicated stresses for 2 hours at which point total protein was extracted. Lysates were subjected to analysis by SDS-PAGE followed by Western Blotting using antisera to either FLAG or GAPDH (loading control). Con, control: Ca, Caffeine; CFW, Calcofluor White; HU, Hydroxyurea; SDS, Sodium Dodecyl Sulfate. (C) Western blot analysis of HIS-Ydj1 abundance from *E. coli* cells transformed with either a control plasmid or a plasmid expressing the indicated Ydj1 mutant. Lysates were probed with antisera to HIS and total protein was assessed by Ponceau staining of the membrane. In A-C examples of 3 biological replicates are shown.

farnesylation and J-domain acetylation are independently regulated processes (Fig 3A). Interestingly, we also noted an apparent increase in size of Ydj1 in K46Q, K48Q and 6KQ mutants (Fig 3A). Overall, these observations demonstrate that even a single K to R mutation can impact Ydj1 abundance and apparent size on SDS-PAGE. To determine whether this apparent increase in size and abundance was dependent on expression in yeast, we constructed plasmids carrying C-terminally His-tagged Ydj1 wild-type and mutants and expressed the proteins in *E. coli*. A single band for Ydj1 was observed after SDS-PAGE followed by Western blot analysis, presumably due to the lack of farnesylation machinery in bacteria. Interestingly, the mobility difference observed between the 6KQ and 6KR mutants was still present in the recombinant system (Fig 3C). A well-established consequence of purifying recombinant Ydj1 is the co-purification of a Ydj1 fragment lacking the J-domain, DJ. After purification of wild-type and mutant recombinant Ydj1 from bacteria, we analyzed Coomassie-stained SDS-PAGE gels for the presence of the DJ fragment. This fragment was present in all samples with a consistent apparent molecular weight, confirming that the size alterations seen in the full-length mutant proteins are due to differences in the J-domain (S1A Fig).

## J-domain acetylation impacts the global Ydj1 interactome

The interactions between chaperones and co-chaperones are modulated by cellular stress and the fine-tuning of stress response through post-translational modifications, also known as the chaperone code [19,32–34]. To determine whether Ydj1 protein interactions are modulated by J-domain acetylation, we quantitatively compared the interactomes of Ydj1-FLAG 6KQ and Ydj1-FLAG 6KR. Ydj1-FLAG complexes from cells expressing the 6KQ or 6KR mutants were isolated via FLAG-Dynabeads. These complexes were digested via trypsin, and peptides were comparatively analyzed using LC-MS (Fig 4A). Protein interactions of Ydj1 were normalized against the bait (Ydj1), and then the log2 interaction ratio (6KQ/6KR) was calculated. A change in the interaction between 6KQ and 6KR samples was considered significant if the normalized log2 (6KQ/6KR) value was >1 or <-1. The interactors were then sorted according to their Gene Ontology terms and were plotted with the y-value corresponding to a change in chaperone interaction (Fig 4B). Three hundred twenty-seven high-confidence Ydj1 interactors were identified, and ~63% of these were unaffected by Ydj1 acetylation (Fig 4B). Consistent with the loss of Ydj1 function seen in the 6KQ mutant, 21% of Ydj1 interactors had a strong preference for the 6KR mutant, with 16% displaying an enhanced affinity for the 6KQ sample (Fig 4B). Gene Ontology (GO) analysis uncovered differences in the classes of proteins enriched for interaction between the 6KR and 6KQ samples (Fig 4B). For example, proteins involved in vesicle trafficking, chromatin regulation, mitochondrial function, and tRNA wobble modification tended to be enriched in 6KR complexes. In contrast, proteins involved in protein turnover, metabolism, and cell polarity were enriched in the 6KQ complexes (Fig 4B). Many of the Ydj1 interactions (38%) identified were proteins involved in protein translation. 13% of these showed a preference for the 6KQ mutant, while 10% of these showed a preference for the 6KR version (Fig 4B).

Ydj1 mediates the majority of its functions through interactions with Hsp70 and Hsp90 chaperones as well as assorted co-chaperone proteins [35]. Out of a total of the 120 acetylation state-dependent Ydj1 interactors identified only 15 (12%) have not been previously identified as a client of the Hsp90 or Hsp70 system in yeast (Fig 4C).

Modeling of the Ssa1-Ydj1 interaction with AlphaFold 3 suggested that all six Ydj1 acetylation sites are in close proximity to the interaction surface with Ssa1 (Fig 4D). The model predicted that several of these sites, K23, K37 and K48 directly interact with the nucleotide-binding domain (NBD) of Ssa1 (Fig 4D). We considered the possibility that the altered

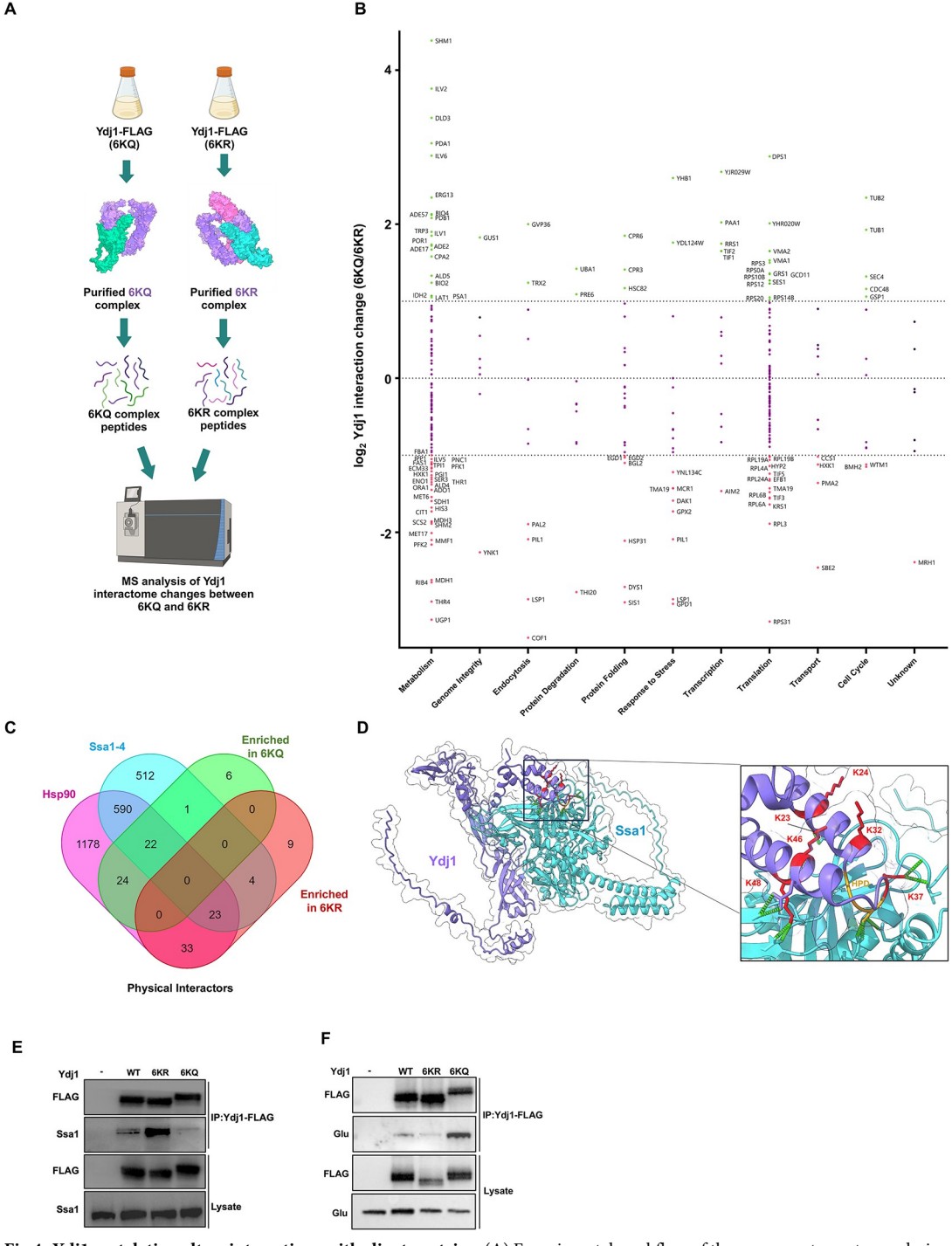

**Fig 4. Ydj1 acetylation alters interactions with client proteins.** (A) Experimental workflow of the mass spectrometry analysis of Ydj1 complexes purified from yeast cells. *ydj1Δ* cells expressing either 6KQ-FLAG or 6KR-FLAG variants of Ydj1 were grown to mid-log phase, and Ydj1 complexes were isolated via FLAG-dynabeads. 6KQ and 6KR complexes were digested into peptides via trypsin and were analyzed by mass spectrometry. Created in BioRender. Mitchem, M. (2024) BioRender.com/e89j574. (B) Comparative interactome analysis of Ydj1 6KQ and 6KR complexes. Interactors were organized into functional categories and plotted against 6KQ/6KR interaction change (Log2 ratio). The dotted lines represent an interaction change of Log2 > 1 or Log2 < −1. Interactors are colored according to change in interaction as follows: green (significant increase), red (significant decrease), or purple (no significant change) 6KQ/6KR. (C) Venn diagram of Ydj1 interactions observed in this experiment vs known physical interactors of Ssa1-4 and Hsp90 (both Hsc82 and Hsp82). Data for Hsp82, Hsc82, Ssa1-4 were obtained from the Saccharomyces Genome Database. (D) Acetylated lysine sites on Ydj1 are located on the interaction surface between Ydj1

and Ssa1. The Ydj1-Ssa1 complex was generated via AlphaFold 3 and rendered in ChimeraX. The six acetylation sites are labeled in red and the conserved HPD region is colored yellow. (E) Acetylation of Ydj1 disrupts interaction with Ssa1. Ydj1 complexes were isolated from the indicated yeast mutants using FLAG-dynabeads, and interaction with Ssa1 was assessed via Western Blotting using antisera to FLAG and Ssa1. (F) Acetylation of Ydj1 enhances interaction with Hsp82. WT, 6KR and 6KQ cells were transformed with a plasmid expressing Glu-tagged Hsc82. Ydj1 complexes were isolated from the indicated yeast mutants using FLAG-dynabeads, and interaction with Hsc82 was assessed via Western Blotting using antisera to FLAG and Glu epitope tags.

interactome observed in Fig 4B may be a result of the altered Ydj1-Ssa1 association. In our interactome study, Ydj1 co-purified with Hsp70 paralogs Ssa1 and 2 and Hsp90 paralogs Hsc82 and Hsp82 (Fig 4B and S2 Table). We also identified important members of the proteostasis network including Cpr6, Sis1, Ssb1/2, Sse1/2, Ssc1, Ssz1, Tsa1, Frp1, Pdi1, Hsp26, Hsp60, Hsp104 and Cpr3 (Fig 4B and S2 Table). While the majority of Ydj1 chaperone and co-chaperone interactions were unchanged between the Ydj1 6KR and 6KQ interactomes, increased interaction was detected between Ydj1 and Hsc82, Cpr6 and Cpr3 in the 6KQ sample. In contrast, the interaction between Ydj1 and the related co-chaperone Sis1 was decreased in the 6KQ sample (Fig 4B).

Proteomic analysis of yeast Hsp70 Ssa chaperones is complicated due to their highly similar amino acid sequences [36]. We performed a direct immunoprecipitation experiment to clarify whether the Ydj1-Ssa1 interaction was truly perturbed by acetylation. Yeast expressing wild-type, 6KQ, or 6KR Ydj1-FLAG were grown to mid-log, the protein was extracted, and Ydj1 complexes were isolated by FLAG-Dynabeads. Immunoprecipitates were probed for the presence of Ssa1 using antisera specific to Ssa1 [37]. An enhanced Ydj1-Ssa1 interaction was observed in the 6KR mutant, while almost no interaction was seen between the two proteins in the 6KQ mutant (Fig 4E). In addition, we also carried out a directed immunoprecipitation experiment to validate acetylation-mediated alteration of the Ssa1-Hsc82 interaction. Yeast expressing wild-type, 6KQ, or 6KR Ydj1-FLAG were transformed with a plasmid expressing Glu-tagged Hsc82. After immunoprecipitation using FLAG dynabeads, there was a clear enrichment of Hsc82 in the 6KQ sample and a subtle decline in the amount of Hsc82 co-precipitated in the 6KR sample compared to WT (Fig 4F). Overall, this data suggests that acetylation may promote the movement of Ydj1 from Ssa1 to Hsc82 during the protein folding process.

## J-domain acetylation of Ydj1 fine-tunes translational fidelity

Chaperones bind newly translated proteins and help them achieve their native, active state [38]. It is thus unsurprising that we identified numerous proteins involved in protein translation in our proteomics screen. In particular, Ydj1 co-purified with proteins involved in tRNA processing, ribosome structure, translational initiation, and elongation (Fig 5A). To understand if there was a spatial specificity for acetylation-dependent Ydj1 interactors, we mapped the identified Ydj1 interactors to the structure of the yeast ribosome (PDB: 4V7R) [39] (Fig 5B). Proteins clustered together in the ribosome's small (40S) subunit tended to prefer interaction with the 6KQ mutant, whereas those in the large (60S) subunit decreased in interaction (Fig 5B). Changes in protein interactions in proteomic studies can be a result of either changes in the strength of association or altered abundance of the interactor. To determine whether ribosomal protein hits in our screen were caused by changes in protein abundance, we utilized the Dharmacon Yeast ZZ-tag ORF collection (https://horizondiscovery.com/en/non-mammalian-research-tools/products/yeast-orf-collection). These plasmids express ZZ-HA--HIS-tagged yeast ORFs from the inducible galactose promoter. We constitutively expressed three ribosomal proteins (Rps3, Rps0 and Gcd11) in wild-type and Ydj1 acetylation mutant

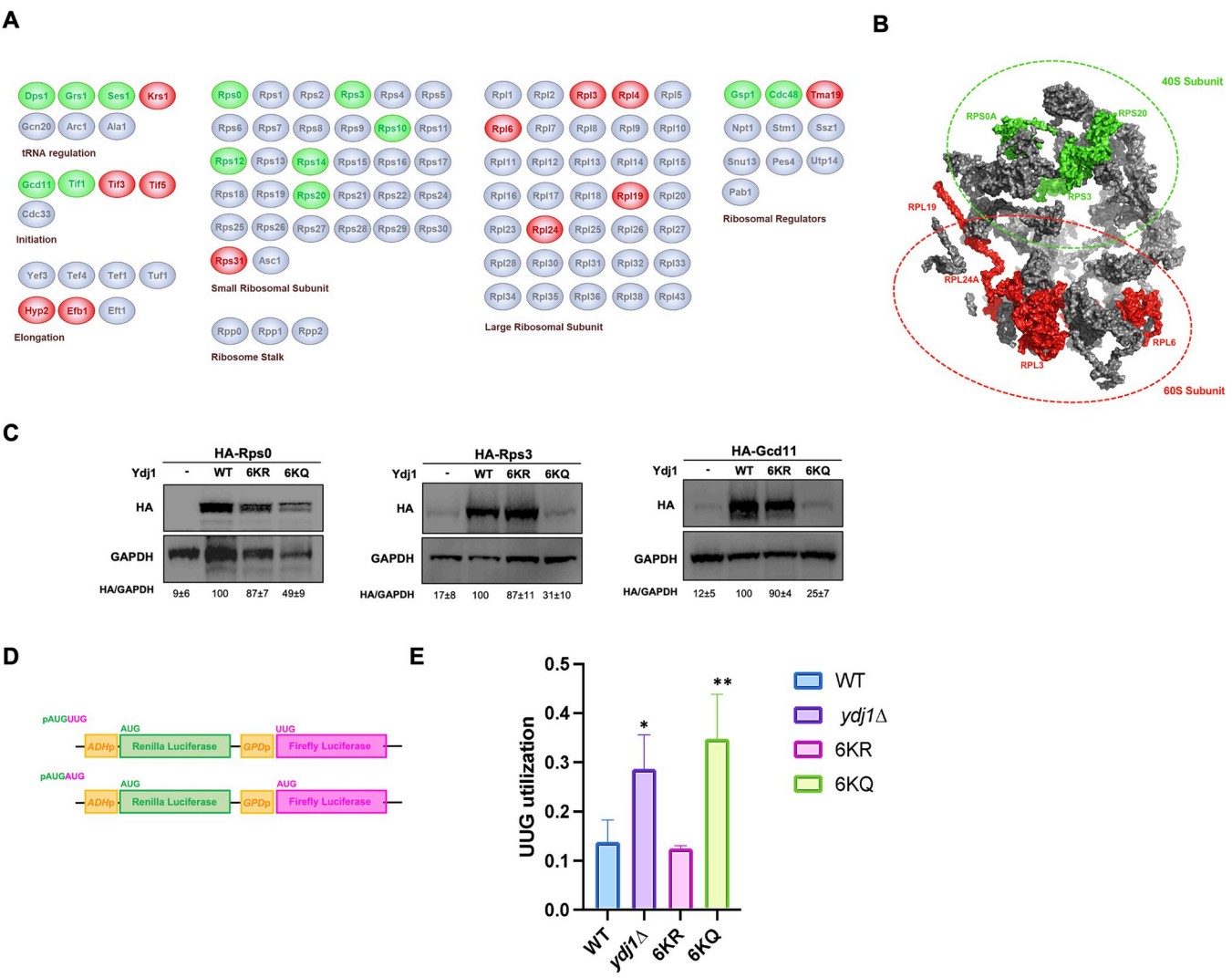

**Fig 5. Ydj1 acetylation impacts multiple aspects of translation.** (A) Interaction of Ydj1 with machinery involved in protein translation is impacted by Ydj1 acetylation. Interactors from the previous MS experiment were grouped by function and colored based on preference for either KR or KQ variant of Ydj1. Green, interactor has a preference for binding the KQ version: Grey, interactions are unaffected by Ydj1 acetylation; Red, interactor has a preference for the KR variant of Ydj1. (B) Ribosomal proteins displaying acetylation-dependent interaction with Ydj1 were mapped to the structure of the yeast ribosome (PDB: 4V7R). Interactors in green are those that increased in interaction, while those in red are interactors that showed a decrease in interaction (6KQ/6KR). (C) Ydj1 function impacts the stability of selected proteins involved in translation. *ydj1Δ* cells expressing plasmids encoding Ydj1 J-domain mutant proteins were transformed with plasmids for the expression of galactose-driven HA-tagged ribosomal proteins. Abundance of ribosomal proteins was assessed by Western blot analysis using antisera to HA and GAPDH. Band intensities were measured using ImageLab (Biorad) which were normalized against the loading control. Each normalized value was then compared to the normalized value from the wild type. (D) Schematic of the translational fidelity assay used. (E) The fidelity of translation initiation in Ydj1 acetylation mutants was determined using the UUG/AUG dual luciferase assay described in [40]. The data shown for the assays and blots are the mean and standard deviation of three biological replicates. Statistical significance was calculated via ANOVA. (*p < 0.05; **p 0.01).

cells (Fig 5C). The stability of Rps0 was dramatically reduced in cells lacking Ydj1 and moderately reduced in 6KQ cells (Fig 5C). Excitingly, the stability of Rps3 and Gcd11 was also clearly dependent on Ydj1 function, with lowered expression of both proteins in cells either lacking Ydj1 or expressing the KQ mutant (Fig 5C). Given these acetylation-dependent changes in the ribosomal proteins, we considered the possibility that acetylation may alter translational fidelity. We assessed translational fidelity in WT and Ydj1 mutant cells using a well-established dual luciferase assay. In this system, *Renilla* luciferase (Rluc) and firefly luciferase (Fluc) are

under the control of separate constitutive promoters [40] (Fig 5D). The Fluc mRNA has a non-traditional start codon (UUG) which is only used when there is a loss of translational fidelity [40]. The Rluc mRNA has a classic AUG start codon and therefore acts as a control for alterations in overall translation. Using this system, we observed a loss of translational fidelity in cells lacking Ydj1, a phenotype that was recapitulated in KQ, but not KR cells (Fig 5E).

## Acetylation impacts the in vitro function of Ydj1

To explore the underlying reason for the loss of Ydj1 function in the 6KQ, K23Q, and K37Q mutants, we performed a series of in vitro assays using His-tagged Ydj1 wild-type and mutant proteins that were expressed in *E. coli* and purified (S1A Fig). Initially, we measured the ability of wild-type Ydj1 and the Ydj1 arginine and glutamine substitution mutants to assist Ssa1 in the refolding of a model substrate, denatured luciferase (Fig 6A). In control experiments in the absence of the partner protein, Ssa1, Ydj1 wild-type, and the Ydj1 mutants were unable to reactivate luciferase (Fig 6A and 6B). In the presence of the partner protein, the Ydj1 6KQ and K37Q were highly compromised for their ability to reactivate luciferase, while K23Q and K23R were slightly defective compared to wild type. In contrast, the 6KR and K37R exhibited ~two-fold enhanced ability to refold luciferase compared to wild-type (Fig 6A and 6B). The results were confirmed using a second preparation each of purified Ydj1 wild-type and K37R (S1B Fig). Together these results suggest that K37 and to a lesser degree K23 are important for client refolding. As another measure of collaboration between Ssa1 and Ydj1, we monitored the stimulation of Ssa1 ATP hydrolysis by Ydj1 wild-type and mutants (Fig 6E). We observed that Ydj1 6KQ was defective in its ability to stimulate Ssa1 ATP hydrolysis and K37Q was partially defective (Fig 6E). Ydj1 6KR and K37R stimulated Ssa1 ATPase to the same extent as wild-type. Both Ydj1 K23Q and K23R mutants also stimulated ATP hydrolysis similarly to wild-type (Fig 6E). In control experiments, there was insignificant ATP hydrolysis by wild-type Ydj1 and all mutants, as expected (Fig 6E). Ydj1 K37 is located next to the conserved HPD motif at residues 34–36, which is known to interact with Hsp70s [13,14], and both K23 and K37 appear to lie on the interface between Ydj1 and Ssa1, based on our model in Fig 4D. To examine the potential for these residues to participate in the direct interaction between Ydj1 and Ssa1, binding between Ssa1 and His-tagged Ydj1 wild-type and mutants was monitored by BioLayer Interferometry (BLI) (Fig 6C and 6D). 6KQ and K37Q were defective in Ssa1 binding, while 6KR and K37R were partially defective compared to wild-type (Fig 6C and 6D). K23Q and K23R bound Ssa1 similarly to wild-type. It is interesting to note the differences between the in vitro and cellular results caused by mutating K23. These differences may be due to the absence of PTMs on Ydj1 or Ssa1 in the *in vitro* setting which may mediate Ydj1-Ssa1 interaction and client folding in a cellular context. It is interesting that despite the K37Q mutant being functionally inactive in vitro, the same mutant in yeast yields a partially functional protein as evidenced by the 6KQ mutant being able to grow on media containing HU and CFW. It should be noted that although the in vitro experiments were carried out in conjunction with only Ssa1, all Ssa isoforms are present in the yeast mutants. This raises an interesting possibility that although K37 acetylation status is critical for interaction with Ssa1, it may not be as important for interaction with Ssa2, 3 and 4. This may also explain the original ambiguous interactome data regarding Ssa1 and Ssa2. To investigate whether the binding surface for the Ydj1 J-domain may be slightly different in the Ssa isoforms, we modeled these interactions with Alphafold 3 (S2A and S2B Fig). These modeled interactions show only a small difference in their interaction surface but indicate a potential shift in hydrogen bonding between the proteins. When evaluating the model using ChimeraX, we observed that the number of hydrogen bonds between Ssa1 and Ydj1 (S2A Fig) is slightly less than Ssa2 and Ydj1

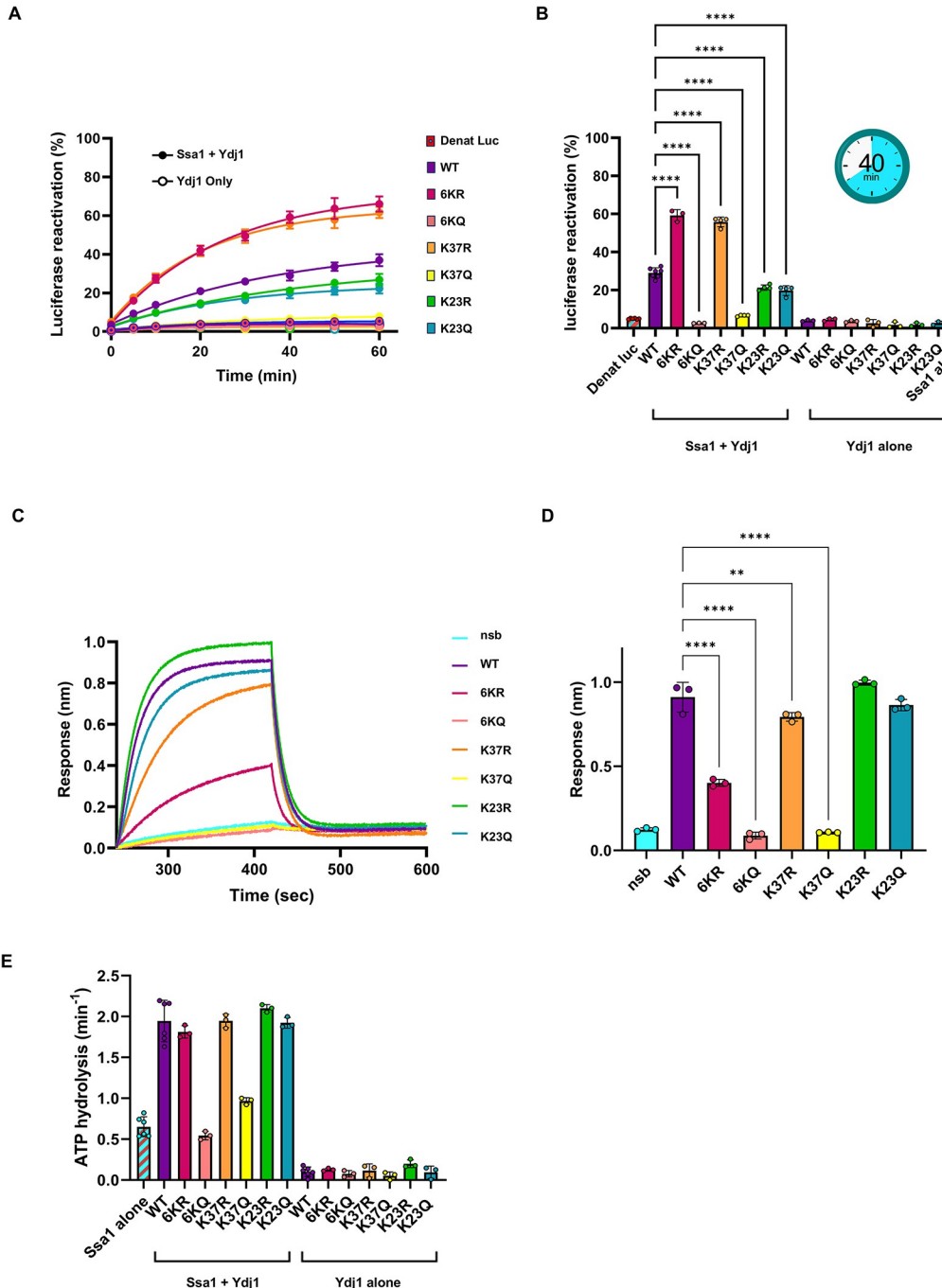

**Fig 6. Acetylation impacts the *in vitro* function of Ydj1.** (A) Reactivation of chemically denatured luciferase was monitored over time in the presence of 3 μM Ssa1 and 0.3 μM Ydj1 wild-type or mutant as described in Materials and Methods. (B) The percentage of luciferase reactivation after 40 min relative to a non-denatured luciferase control is shown with additional controls and data from Fig 6A. (C) The interaction of Ssa1 with Ydj1 wild-type or Ydj1 mutant proteins was analyzed using BioLayer Interferometry as described in Materials and Methods. Biotinylated Ydj1 wild-type or mutant was loaded onto streptavidin biosensors to equivalent levels and the association and dissociation of Ssa1 was monitored. The amount of non-specific Ssa1 binding (nsb) was determined in the absence of Ydj1 on the biosensor. The mean from three independent experiments is shown. (D) Bar graph showing the amount of Ssa1 binding to Ydj1 wild-type or mutant observed in (C) at the conclusion of the association phase. Each Ydj1 mutant is compared to wild-type using one-way Anova, n = 3, **P = 0.007, ****P < 0.0001. (E) The rate of Ssa1 ATP hydrolysis was determined in the absence and presence of Ydj1 wild-type or Ydj1 mutants as described in Materials and Methods. Background ATPase from the individual Ydj1 proteins is also shown. In A, B, D, and E, results are shown as means ± SD of three or more replicates. All shown as mean ± SD, (**p <0.01, ****p <0.0001).

(S2B Fig). When acetylation of K37 on Ydj1 is modeled, Ssa1 shows a larger decrease in hydrogen bonding compared to the WT Ydj1 than is seen in the Ssa2 model (S2B Fig). To understand these potential differences further we aligned the amino acid sequences of Ssa1-4 and examined this for sequence variation on sites predicted to be important in interacting with Ydj1 (S2C Fig). There was a lack of conservation of the Ssa isoforms around residues 190, 210, 380 and 480 (Ssa1 numbering), sites important for Ydj1 interaction (S2C Fig). Taken together, these models suggest a potential difference in Ssa-Ydj1 interaction among the Ssa isoforms.

## Discussion

Over the past decade, research has unveiled a "Chaperone Code"—a system of post-translational modifications (PTMs) that regulate molecular chaperone function [32,33,41,42]. Despite the identification of thirty PTM sites on Ydj1, their specific roles and regulatory mechanisms remain largely unknown. Lysine acetylation is a rapidly reversible PTM tightly controlled through lysine acetyltransferases (KATs) and lysine deacetylases (KDACs) [23,43]. In *Saccharomyces cerevisiae*, around 4000 lysine acetylation sites have been uncovered on proteins involved in DNA repair, chromatin remodeling, cellular metabolism, protein folding, and transcription (via histone acetylation/deacetylation) [43,44]. In this study, we set out to determine the relevance of Ydj1 acetylation. We selected six previously identified acetylation sites that were highly conserved and present in the J-domain of Ydj1, a region critical for Ydj1 function. Mutation of these sites to either prevent or mimic acetylation at these sites produced an interesting spectrum of phenotypes. Surprisingly, prevention of acetylation produced no detectable effect. In contrast, K23Q, K37Q, and 6KQ showed a complete loss of growth at high temperatures. The J-domain of Ydj1 forms an interaction surface that allows the HPD region to bind to and stimulate the ATPase activity of Ssa1, thereby closing Ssa1's lid and trapping the client protein [2]. This activity is highly regulated, though there is still some debate as to the current mechanisms behind this. In our AlphaFold generated model of the Ssa1-Ydj1 complex, K23 and K37 lie directly on the interaction surface and participate in hydrogen bonding between Ydj1 and Ssa1. The K37Q yeast mutant displays the most severe phenotype of the individual mutants which appears to be corroborated by *in vitro* data clearly demonstrating that Ydj1 K37Q is severely defective in Ssa1 binding, stimulation of Ssa1 ATPase activity and refolding of a model client protein. Mutation of K23 however produced a more subtle phenotype. Although the K23Q yeast mutant is temperature sensitive and displays a reduced ability to refold luciferase *in vitro*, it can still interact with Ssa1 and stimulate ATP hydrolysis. These differences may be a result of PTMs that are present on Ssa1 and Ydj1 in a cellular environment, but which are absent in recombinant protein assays. The 6KQ yeast mutant displays almost a complete loss of Ydj1 activity, as evidenced by sensitivity to all the stresses tested. Similar to the K37Q mutant, the 6KQ is unable to bind Ssa1, stimulate ATP hydrolysis or promote client refolding. Taken together, these results suggest that a significant proportion of the phenotypes seen in the KQ mutants are likely due to impaired Ydj1-Ssa1 interaction. Interestingly, our *in silico* modeling of the acetylated Ydj1 dimer suggests that the 6 acetylation sites may promote a J-domain rearrangement and enhanced Ydj1-Ydj1 protomer interaction. While clearly this model needs future experimental validation, if correct it may go some way to explain the observed loss of Ydj1 function in 6KQ. Given the potential disruption of Ydj1-Ssa1 interaction by acetylation, we anticipated that the acetylation of Ydj1 would have far-reaching effects on its global protein interactions. In our MS experiment, we identified 327 interactors of Ydj1, 63% of which were unaffected by the acetylation status of Ydj1, and 16% of Ydj1 interactions were enriched in the KQ, suggesting that acetylation subtly fine-tunes Ydj1 interactions. Despite structural evidence as well as *in vitro* and *in vivo* assays showing a clear impact

of acetylation on the Ssa1-Ydj1 interaction, this loss was not recapitulated in our mass spectrometry analysis. Our explanation for this discrepancy lies in the challenges of distinguishing the highly similar Ssa isoforms Ssa1, 2, 3, and 4 by mass spectrometry. To bypass this issue, we performed a co-immunoprecipitation/Western Blot analysis of the Ydj1-Ssa1 interaction using antisera specific to the Ssa1 isoform. Consistent with our original hypothesis, acetylation of Ydj1 indeed disrupted interaction with Ssa1. Interestingly, the 6KR mutant displayed a higher-than-normal Ssa1-Ydj1 interaction, which may suggest that wild-type Ydj1 has a low, but non-zero level of acetylation at these sites.

The physiological relevance of yeast possessing four nearly identical Ssa proteins has not been fully explored [36]. It is possible as with Hsp90 isoforms that Ssa1-4 possess slightly different client and co-chaperone binding specificities [45]. Our modeling suggests that there is amino acid variance between Ssa1, 2, 3 and 4 at the region that binds Ydj1 and thus may generate a differential response to Ydj1 acetylation. While beyond the scope of this study, this may explain the ambiguous Ssa proteomic data and how K37Q can still function in yeast while being inactive in in vitro assays with Ssa1.

The current model of client processing is the client is first bound by Hsp40 and then delivered to Hsp70. This, in turn, activates the ATPase activity of Hsp70, and both Hsp70 and Hsp40 interact with the middle domain of Hsp90. Hsp70 and Hsp40 eventually dissociate from this transient complex, leaving the client with Hsp90 for the final stages of client maturation [10]. The driving forces behind client transfer from Hsp70 to Hsp90 remain obscure. It has been proposed in mammalian cells that the NudC protein is a key mediator of chaperone-client transfer, but the protein is not present in yeast [46]. Data from our MS and IP experiments suggest that Ydj1 acetylation disrupts the Ydj1-Ssa1 interaction while strengthening the Ydj1-Hsc82 interaction. Overall, this raises an interesting possibility that Ydj1 acetylation may be a key component of the protein folding process, directing Ydj1 between Ssa1 and Hsc82.

Our proteomic analysis uncovered a large number of Ydj1 interactors associated with protein translation, in line with several previous studies that have hinted at a connection between Ydj1 and protein synthesis. For example, the temperature-sensitive Ydj1 mutant *ydj1-151*, displays defects in the translation of two heterologously expressed proteins, firefly luciferase and GFP [47]. Additionally, cells lacking Ydj1 are sensitive to inhibitors of translation such as hygromycin B and cycloheximide [28]. Protein synthesis is carried out by the ribosome, an RNA-protein complex consisting of a small and large subunit [48]. Ribosome assembly involves more than 200 assembly factors, 79 ribosomal proteins, rRNAs, and other ribosomal-related proteins that participate in the complex pathway to achieve protein synthesis [48,49]. Assembly of the ribosome begins with transcription of rRNA in the nucleolus, then the rRNA undergoes complex folding, nucleotide modification, and binding to ribosomal proteins [49]. In *S. cerevisiae*, the small subunit (40S) has 33 ribosomal proteins and an 18S ribosomal RNA [50]. The large subunit (60S) has 46 ribosomal proteins and three ribosomal RNAs [48]. The small subunit is the location of the decoding site, where the anticodon of an amino-acyl tRNA base pairs with its respective codon found in the mRNA [48]. The large subunit contains the peptidyl transferase center (PTC), which is the active site of the unit. This is where rRNA catalyzes the formation of peptide bonds and hydrolyzes the peptidyl-tRNA bond [48]. Many of the interactors of Ydj1 identified in our screen were structural components of the ribosome, both from the small and large subunit. Recent studies have revealed that Hsp90 is primarily associated with the small subunit, while Hsp70 (which binds the nascent chain) was found to be associated with all the ribosome fractions [51]. Rps0, Rps3, and Gcd11 were not only identified as Ydj1 interactors, but their abundance was drastically reduced in cells lacking wild-type Ydj1 and expressing 6KQ. Overall, this suggests that these proteins are novel *bona fide* clients of Ydj1. Rps0, Rps3 and Gcd11 are destabilized in the 6KQ yeast despite displaying an

enhanced interaction with Ydj1-6KQ in our proteomic experiment. We have observed a similar phenomenon previously, for example Ssa1 T36 phosphorylation increases Ssa1-Cln3 cyclin interaction to promote Cln3 degradation as part of cell cycle progression and the nutrient starvation response [52]. Lysine acetylation of Ydj1 may be part of the process for cells to decide whether to fold a client or target it for degradation. Alternatively, a locked client-Ydj1 complex may prevent clients from being passed to downstream folding machinery such as Ssa1 in the first place leading to client degradation.

Several PTMs, such as lysine ubiquitination, play an important role in the regulation of protein stability, such as the rise and fall of cyclins throughout the cell cycle. The K37Q mutant displays a higher abundance than wild-type Ydj1, and as our mutants were expressed from a constitutive promoter, this change in abundance is likely caused by altered protein degradation. We have considered the possibility that K37 may also be a site of ubiquitination or SUMOylation. In this scenario, acetylation would prevent these modifications, leading to a stabilization of Ydj1; however, the levels of Ydj1 in the K37R mutant do not corroborate this theory. It is, of course, possible that K37Q prevents ubiquitination at a lysine not studied in this project, which will be investigated further in the future. Farnesylation of Ydj1, which can be detected as a doublet on Western Blots, can be clearly seen in all mutant samples. This suggests that while acetylation may impact association with Ssa1, farnesylation of Ydj1 remains unaffected.

Overall, our results suggest that acetylation of the residues located in the J-domain (particularly K23 and K37) inhibit Ydj1 function. Our data thus produce a conundrum: Why would the cell want to inhibit Ydj1 activity? The G1 cyclin Cln3 competes with Ydj1 for binding to Ssa1 during cell cycle progression and nutrient starvation [53]. T36 phosphorylation of Ssa1 promotes displacement of Ydj1 from Ssa1 and allows Cln3 to bind, triggering Cln3 destruction [52]. It is thus possible that Ydj1 acetylation is part of the same mechanism to regulate Ssa1-Ydj1 interaction during cell cycle progression. Similarly, our lab has shown that deacetylation of Ssa1 is needed for a robust heat shock response [27]. This evidence, combined with our data, suggests that deacetylation of both Ssa1 and Ydj1 may be coordinated to increase Ssa1-Ydj1 interaction and trigger refolding of important client complexes under stressful conditions. We do acknowledge a limitation of this this study is the use of mutations to prevent and mimic acetylation. While KR and KQ mutations are extremely common ways to study the impact of lysine acetylation, it is possible that these mutations may alter the surrounding landscape of PTMs or alter Ydj1 structure in unanticipated ways. While beyond the scope of this initial study, going forward we hope to mimic acetylation *in vitro* and *in vivo* through direct incorporation of acetyl-lysine using genetic code expansion technologies.

Taken together, this study provides evidence that the J-domain acetylation of Ydj1 has a significant impact on its chaperone and client interactions, leading to altered stress resistance. While beyond the scope of this study, we envisage future work that explores the conditions and enzymes that control Ydj1 acetylation. HDAC6 deacetylates the Ydj1 homolog DNAJA1 in mammalian cells, a process required for Hsp70-DNAJA1 interaction [23]. Although the specific sites of acetylation on DNAJA1 have not yet been identified, it is possible from the evidence presented in this study that K23 and K37 equivalent sites on DNAJA1 are responsible. While these studies of the chaperone code are still in their infancy, it will be exciting to explore not only the role and regulation of chaperone and client PTMs but also their interplay in proteostasis.

## Experimental procedures

**Reagents and resources.**   Details on all reagents and resources (yeast strains and plasmids) are provided in S1 Table.

**Yeast Strains and growth conditions.** Yeast cultures were grown in either YPD (1% yeast extract, 2% glucose, 2% peptone) or grown in SD (0.67% yeast nitrogen base without amino acids and carbohydrates, 2% glucose) supplemented with the appropriate nutrients to select for plasmids and tagged genes. *Escherichia coli* DH5α was used to propagate all plasmids. *E. coli* cells were cultured in Luria broth medium (1% Bacto tryptone, 0.5% Bacto yeast extract, 1% NaCl) and transformed to ampicillin or kanamycin resistance by standard methods. For monitoring stress conditions, cells were grown to mid-log phase, 10-fold serially diluted, and then plated onto appropriate media using a 48-pin replica-plating tool. Images of plates were taken after three days at 30˚C. For experiments to determine the impact of stress on abundance of Ydj1-6KQ, cells were grown to early mid-log phase in SD-HIS media at 30˚C. Cells were then treated with the following conditions where indicated; 37˚C, 10mM caffeine, 50μg/ml Calcofluor White, 200mM hydroxyurea, 0.02% Sodium Dodecyl Sulphate (each for 2 hours) or 100J/m$^2$ UV.

**Plasmid construction.** The plasmid for expressing Ydj1 with a C-terminal FLAG tag (pYCP-GPD-YDJ1-FLAG) was constructed by VectorBuilder, and acetylation site mutants were generated by Genscript. Plasmids are listed in supplementary data (S1 Table), and full plasmid sequences in Snapgene format are available on request. For bacterial expression, the *YDJ1* gene was PCR amplified from yeast genomic DNA to produce overhangs containing XbaI and XhoI restriction sites on the 5' and 3' ends of the PCR product. After digestion with XbaI and XhoI, the YDJ1-containing fragment was inserted into *E. coli* expression plasmid pRSETA (Thermo) resulting in a construct that allows expression of a 6xHIS-TEV-Ydj1 fusion protein.

**Expression of recombinant Ydj1 in *E. coli*.** HIS-Ydj1 expression plasmids were transformed into *E. coli* BL21-competent cells. Cells were grown to an OD$_{600}$ = 0.6 in 2YT media supplemented with 100 μg/ml ampicillin at 37˚C. Once the appropriate OD was reached, protein expression was induced by adding IPTG at a 1 mM final concentration. Cultures were incubated at 37˚C with shaking for an additional 4 hrs. Induced cells were then harvested via centrifugation (5000 rpm, 5 min) and sonicated in intervals of 30 seconds on ice until clear. Protein extracts were stored at -20˚C until prepared for Western Blot analysis.

**Immunoprecipitation.** For FLAG IP, cells were harvested, and FLAG-tagged proteins were isolated as follows: Protein was extracted via bead beating in 500 μl binding buffer (50 mM Na-phosphate pH 8.0, 300 mM NaCl, 0.01% Tween-20). 200 μg of protein extract was incubated with 30 μl anti-Flag M2 magnetic beads (Sigma) at 4˚C overnight. Anti-FLAG M2 beads were collected by the magnet and then washed 5 times with 500 μl binding buffer. After the final wash, the buffer was aspirated, and beads were incubated with 65 μl Elution buffer (binding buffer supplemented with 10 μg/ml 3X FLAG peptide (Apex Bio) for 1 hr at room temperature, then beads were collected via magnet. The supernatant containing purified FLAG-protein was transferred to a fresh tube, 25 μl of 4X SDS-PAGE sample buffer was added, and the sample was denatured for 5 min at 95˚C. The eluates were separated by SDS-PAGE (7.5–10%) and visualized by immunoblotting.

**Immunoblotting.** Protein extracts were made as described in [22]. 20 μg of protein was separated by 4%–12% NuPAGE SDS-PAGE (Thermo Fisher Scientific). Proteins were detected using the primary antibodies (S1 Table) with the following dilutions: anti-GAPDH (1:5000), anti-HA-tag (1:2000), anti-FLAG tag (1:2000), anti-His tag (1,2000), anti-Ssa1 (1:2000), anti-Glu tag (1,2000) and anti-PGK1 (1:5000). Membranes were washed with TBS-Tween 20 (0.1%) and incubated with the corresponding secondary antibodies: anti-rat IgG-HRP (1:5000), anti-mouse IgG-HRP (1:5000), anti-rabbit IgG-HRP (1,5000), and anti-mouse IgM-HRP (1,5000). Blots were imaged on a ChemiDoc MP imaging system (Bio-Rad). After treatment with SuperSignal West Pico Chemiluminescent Substrate (Thermo Fisher

Scientific). Blots were stripped and re-probed with the relevant antibodies using Restore Western Blot Stripping Buffer (Thermo Fisher Scientific).

**Quantitation of Western Blots.** The blots were quantitated using Biorad Image Lab software. Data was the average and standard deviation from three replicates. Levels of ribosomal proteins- Gcd11, Rps0 and Rps3 in WT or ydj1Δ or mutant cells were determined by western blotting. Relative protein expression was measured based on band intensities obtained from ImageLab (Biorad) which were normalized against that loading control. Each normalized value was then compared to the normalized value from the wild type.

**Recombinant Protein Purification.** Ssa1 was isolated as previously described [54]. His-tagged Ydj1 wild-type and mutant proteins were expressed in *E. coli* Rosetta (DE3) cells in the presence of IPTG. His-tagged Ydj1 proteins were purified using Ni-sepharose (Cytiva) in Ni buffer (40 mM Hepes (pH 7.5), 0.3 M NaCl, 10% glycerol(vol/vol)). Bound protein was washed with Ni buffer containing 10 mM imidazole followed by 25 mM imidazole and then eluted with Ni-buffer containing 250 mM imidazole. Ydj1 containing Ni-sepharose fractions were pooled and chromatographed on a HiLoad Superdex 200 size exclusion column in 25 mM Hepes (pH 7.5), 0.1 M NaCl, 0.1 mM EDTA, 1 mM DTT, 5% glycerol (vol/vol) using an AKTA pure 25 system. For BLI experiments, Yd1 proteins were labeled using a 1.5-fold excess of NHS-PEG4-Biotin (Thermo Scientific, Life Technologies). Excess biotin reagent was removed using 7K molecular weight cut-off Zeba Spin Desalting Columns (Thermo Scientific). Concentrations given are for Ydj1 dimers and for Ssa1 and luciferase monomers.

**Biolayer interferometry (BLI) analysis.** BLI was used to monitor the interaction between Ssa1 and Ydj1 wild-type or mutant proteins using a Sartorius Octet R4 instrument and streptavidin biosensors at 23˚C. Wild-type Ydj1-biotin (5 μg/ml) was loaded on the biosensors and the association of Ssa1 (1 μM) with Ydj1 was monitored over time followed by dissociation in the absence of Ssa1. Interaction of Ssa1 with Ydj1 mutant proteins was determined using a loading of Ydj1-biotin mutant proteins equivalent to Ydj1-biotin wild-type. All Ydj1, Ssa1 BLI steps were performed in 25 mM Hepes pH 7.5, 50 mM KCl, 2 mM DTT, 5 mM $MgCl_2$, 0.5 mM ATP and 0.02% Tween-20 (vol/vol). Ssa1 nonspecific binding was monitored using a reference biosensor subjected to each of the above steps in the absence of the biotinylated Ydj1, and the nonspecific binding signal was subtracted from the corresponding experiment.

**Luciferase reactivation assay.** Luciferase (8 μM) was chemically denatured in 5 M guanidine hydrochloride, 25 mM Hepes pH 7.5, 50 mM KCl, 0.5 mM EDTA, for 10 min at 23˚C. For reactivation, denatured luciferase was diluted 100-fold into 25 mM Hepes pH 7.5, 0.1 M KOAc, 5 mM DTT, 10 mM Mg(OAc)$_2$, 2 mM ATP, an ATP regenerating system (10 mM creatine phosphate, 3 μg creatine kinase), 3 μM Ssa1 and 0.3 μM Ydj1 wild-type or mutant. Aliquots were removed at the indicated times, and light output was measured using a Tecan Spark in luminescence mode with an integration time of 1000 ms following the injection of luciferin (50 μg/ml). Reactivation was determined compared to a non-denatured luciferase control.

**ATPase assay.** Steady-state ATP hydrolysis was measured at 37˚C in 25 mM Hepes pH 7.5, 50 mM KCl, 2 mM DTT, 0.01% (vol/vol) Triton X-100, 5 mM $MgCl_2$, and 2 mM ATP using a pyruvate kinase/lactate dehydrogenase enzyme-coupled assay as described [55] and 1 μM Ssa1 and 1 μM Ydj1 wild-type or mutant.

## Mass Spectrometry

**In-solution sample digestion and desalting.** Ydj1 complex immunoprecipitates were eluted from beads using 8M Urea,10mM DTT in 50 mM Tris pH 8.5 for 45 min at room temperature with mixing at 600 rpm before alkylation with 50 mM IAA for 30 min in the dark.

Samples were diluted 6x with 50 mM Tris pH 8.5 to reach a 2 M Urea concentration and digested with 0.4 µg of trypsin-LysC mix (Promega) overnight at 37˚C. Tryptic peptides were desalted with Pierce C18 Desalting Spin Columns (Thermo Fisher Scientific) according to the manufacturer's protocol, dried down on SpeedVac, and resuspended in mobile phase A (0.2% formic acid in water) immediately prior to mass spectrometric analysis.

**Liquid chromatography-tandem mass spectrometry peptide analysis.** Resuspended peptides were separated by nanoflow reversed-phase liquid chromatography (LC). An Ultimate 3000 UHPLC (Thermo Scientific) was used to load ~1 µg of peptides on the column and separate them at a flow rate of 300 nL/min. The column was a 15 cm long EASY-Spray C18 (packed with 2 µm PepMap C18 particles, 75 µm i.d., Thermo Scientific). The analytical gradient was performed by increasing the relative concentration of mobile phase B (0.2% formic acid, 4.8% water in acetonitrile) in the following steps: from 2% to 30% in 32 min, from 30% to 50% in 5 min, and from 50 to 85% in 5 min (for washing the column). The 4 min wash at high organic concentration was followed by moving to 15% in 2 minutes, increasing to 70% in 1 min for a secondary wash before re-equilibration of the column at 2% B for 7.5 min, for a total run time of 68 min. A 2.2 kV potential was applied to the column outlet using an in-house nanoESI source based on the University of Washington design for generating nano-electrospray. All mass spectrometry (MS) measurements were performed on a tribrid Orbitrap Eclipse (Thermo Scientific). Broadband mass spectra (MS1) were recorded in the Orbitrap over a 375–1500 m/z window, using a resolving power of 120,000 (at 200 m/z) and an automatic gain control (AGC) target of 4e5 charges (maximum injection time: 50 ms). Precursor ions were quadrupole selected (isolation window: 1.6 m/z) based on a data-dependent logic, using a maximum duty cycle time of 3 s. Monoisotopic precursor selection and dynamic exclusion (60 s) were applied. Peptides were filtered by the intensity and charge state, allowing the fragmentation only of precursors from 2+ to 7+. Tandem mass spectrometry (MS2) was performed by fragmenting each precursor passing the selection criteria using both higher energy collisional dissociation (HCD) with normalized collision energy (NCE) set at 30% and electron transfer dissociation–higher energy collisional dissociation (EThcD), with ETD reagent target set at 5e5, reaction time calculated on the basis of a calibration curve and supplemental collisional activation set at NCE = 10%. The AGC target for both HCD and EThcD MS2 was set at 8e4 (maximum injection time: 55 ms), and spectra were recorded at 15,000 resolving power.

**MS Data analysis.** For general identification of all proteins included in the samples, HCD fragmentation data were processed with Protein Discoverer 2.4 utilizing Sequest HT and MS Amanda search engines. For both Precursor Mass Tolerance was 10 ppm and Fragment Mass Tolerance was 0.2 Da. Carbamidomethylation (C) was allowed as a static modification, and dynamic modifications were as follows: Oxidation(M), Acetyl (protein N-term), and Acetyl (K). Identified peptides were validated using Percolator, and the target FDR value was set to 0.01 (strict) and 0.05 (relaxed). Finally, results were filtered for high-confidence peptides using consensus steps. A control peptide error rate strategy was used, and 0.01 (strict) and 0.05 (relaxed) values were used for Target FDR for both PSM and Peptide levels. Changes in protein abundance between the two strains were statistically tested by ANOVA using the built-in function within Proteome Discoverer. The raw proteomic data can be found on the ProteomeXchange #PXD051428 (https://proteomecentral.proteomexchange.org/cgi/GetDataset?ID=PXD051428).

**Structural Modeling.** All structures were obtained by retrieving the FASTA files for each protein from Uniprot: Ydj1 (P25491), Ssa1 (P10591), then uploading them to the Alphafold Server where we either selected for dimerized or monomeric versions of the protein, with or without our targeted residues being acetylated [56]. The .CIF file that was produced from this

software was then imported into ChimeraX, where the models were further depicted for clarity and understanding [57]. All PDB files created are available upon request.

**Amino acid alignment.** Amino acid sequences were obtained for various Ydj1 homologs, including Zebrafish (Q803K1), Northern Mallard (U3J822), Grey short-haired opossum (F6Z1Q5), Human (P31689), Chimpanzee (G2HHH7), Cat (A0A212U399), Mouse (P63037) and Eastern Brown Snake (A0A670Z2W5) using Uniprot. Each sequence was then aligned using Jalview (Fig 1) [58]. Each of the four yeast Ssa proteins (P10591, P10592, P09435 and P22202, sequentially) were also aligned this way.

**Gene ontology enrichment analysis.** Gene ontology analysis of Ydj1 immunoprecipitated complexes was accomplished using GO-SLIM on the Saccharomyces Genome Database.

**General data analyses.** Data processing and analyses were performed using GraphPad Prism (version 7).

## Supporting information

**S1 Fig.** (A) SDS PAGE of purified Ydj1 wild-type and mutants. 0.5 μg of purified His-tagged Ydj1 wild-type and mutant proteins were run on a 4–12% NuPAGE gel (Invitrogen, Thermo Fisher Scientific) using MOPS running buffer. 0.5 μg of purified untagged Ydj1 wild-type was included for comparison. The gel was imaged following staining in InstantBlue Coomassie Protein Stain (Abcam). Molecular weight marker sizes (kDa) are indicated. (B) Reactivation of chemically denatured luciferase by Ssa1 and Ydj1 wild-type or K37R using a second preparation of His-tagged Ydj1 wild-type or mutant. Assays were performed as in Fig 6A. Results are shown as means ± SD of three replicates.
(TIFF)

**S2 Fig.** (A) Predicted interaction surface between Ssa1 and K37 acetylated Ssa1 with the Ydj1 J-domain. The interaction surface is highlighted (magenta), along with respective hydrogen bonding (lime green hash marks). Structures were obtained via the Alphafold 3 Server and mapped and characterized using ChimeraX. (B) Predicted interaction surface between Ssa2 and K37 acetylated Ssa2 with the Ydj1 J-domain. The interaction surface is highlighted (magenta), along with respective hydrogen bonding (lime green hash marks). Structures were obtained via the Alphafold 3 Server and mapped and characterized using ChimeraX. (A) Ssa1-4 Amino acid alignment and consensus graph as determined by Jalview, the red boxes indicate sites of interaction on Ssa1 with WT Ydj1 modeled with AlphaFold 3.
(TIF)

**S1 Table. List of Yeast strains and plasmids used in this study.**
(XLSX)

**S2 Table. List of Ydj1 interactors impacted by Ydj1 acetylation and known physical interactors of Hsp82/Hsc82, Ssa1, 2, 3 and 4.**
(XLSX)

## Acknowledgments

We acknowledge the PRIDE team for the deposition of our data to the ProteomeXchange consortium. Molecular graphics and analyses performed with UCSF ChimeraX, developed by the Resource for Biocomputing, Visualization, and Informatics at the University of California, San Francisco and the Office of Cyber Infrastructure and Computational Biology, National Institute of Allergy and Infectious Diseases.

## Author Contributions

**Conceptualization:** Sue Wickner, Andrew W. Truman.

**Data curation:** Siddhi Omkar.

**Formal analysis:** Siddhi Omkar, Andrew W. Truman.

**Investigation:** Siddhi Omkar, Joel R. Hoskins, Courtney Shrader, Jake T. Kline, Nitika, Luca Fornelli.

**Methodology:** Siddhi Omkar, Joel R. Hoskins, Courtney Shrader, Jake T. Kline, Andrew W. Truman.

**Resources:** Andrew W. Truman.

**Supervision:** Andrew W. Truman.

**Visualization:** Siddhi Omkar, Megan M. Mitchem, Joel R. Hoskins, Courtney Shrader, Jake T. Kline, Luca Fornelli, Sue Wickner, Andrew W. Truman.

**Writing – original draft:** Siddhi Omkar, Megan M. Mitchem, Joel R. Hoskins, Courtney Shrader, Jake T. Kline, Luca Fornelli, Sue Wickner, Andrew W. Truman.

**Writing – review & editing:** Siddhi Omkar, Megan M. Mitchem, Sue Wickner, Andrew W. Truman.

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
