## [Decision Letter · Decision Letter 0]

22 Aug 2024

Dear Dr Truman,

Thank you very much for submitting your Research Article entitled 'Acetylation of the yeast Hsp40 chaperone protein Ydj1 fine-tunes proteostasis and translational fidelity' to PLOS Genetics.

The manuscript was fully evaluated at the editorial level and by independent peer reviewers. The reviewers appreciated the attention to an important problem, but raised some substantial concerns about the current manuscript. Based on the reviews, we will not be able to accept this version of the manuscript, but we would be willing to review a much-revised version. We cannot, of course, promise publication at that time.

If you decide to revise the manuscript for further consideration at PLOS Genetics, please aim to resubmit within the next 60 days, unless it will take extra time to address the concerns of the reviewers, in which case we would appreciate an expected resubmission date by email to plosgenetics@plos.org.

To resubmit, log into your Editorial Manager account and select the option 'Revise Submission' in the 'Submissions Needing Revision' folder.

We are sorry that we cannot be more positive about your manuscript at this stage. Please do not hesitate to contact us if you have any concerns or questions.

Yours sincerely,

Guang-Chao Chen

Academic Editor

PLOS Genetics

Fengwei Yu

Section Editor

PLOS Genetics

The three reviewers raised several questions and concerns regarding the current version. Please carefully consider the reviewers' suggestions and address all their comments with extensive additional experimental support.

Reviewer's Responses to Questions

**Comments to the Authors:**

Reviewer #1: The Chaperone Code describes the complex post-translational regulation of molecular chaperone proteins that confers selectivity and specificity to chaperone function and chaperone-substrate interactions. High-throughput studies have identified numerous chaperone modifications, however functional information is lacking. Here, the authors have made significant progress in understanding lysine acetylation on the major Hsp40 chaperone Ydj1 in yeast. While the manuscript contains some strong and compelling data, additional work is required to support the major claims of the study.

Major Comments:

Despite the framing of this paper around Ydj1 acetylation, the authors have not experimentally validated Ydj1 acetylation at any point in the manuscript. This is absolutely essential to support the interpretation of the results. Alternatively, the authors could express an Isolated J-domain construct containing the mutants to examine domain-specific acetylation in the presence and absence of the Lys mutations. Similarly, it would be fairly straightforward to simultaneously evaluate potential ubiquitination at these residues to address the potential for the mutations to interfere with protein degradation, as mentioned by the authors.

The authors state in the results section that Rps0A levels are moderately reduced in cells expressing the 6KQ mutant. However, this is unclear from the presented data. It is necessary to replace the Rsp0A blot in Fig 5C with a lower exposure to appropriately evaluate the expression levels.

The authors have consistently used the 6KR/6KQ mutants throughout the manuscript, however much of the change in activity is derived from mutation of K23 or K37. Perhaps a double mutant of these two residues would be more representative of Ydj1 acetylation state, and mitigate the potential effects of mutations at the other 4 sites. A more thorough exploration of different mutation combinations is necessary.

Similarly, it is unclear why the authors did not test binding of the 6KQ and 6KR mutants in Fig 6C. This would seem to be essential for correlation with the rest of the data in the manuscript.

In the manuscript the authors have concluded that “the main phenotypic defects seen in the KQ mutants are likely a result of the loss of ability to bind Ssa1 and stimulate the Ssa1 ATPase activity.” However, the authors have performed an interactome and identified many proteins with differential binding to Ydj1 acetylation mutants. Though the authors state that Ydj1 acts through Hsp70/Hsp90, have the authors considered direct effects of Ydj1 activity? In order to support this claim, have the authors determined whether the reported interactors identified in Fig 4B known to rely on Hsp70/Hsp90 chaperoning? If the authors would like to support this conjecture, it would be useful to IP Ssa1 from yeast expressing Ydj1 mutants and repeat the global proteomics, in addition to the interactome of Ydj1 itself. In fact, the observation that the 68% of Ydj1 interactions were not disrupted by acetylation mutants could suggest that these changes are actually independent of Ydj1 regulation of Ssa1.

Related to the previous point, in order to tease out the impact of Ydj1 acetylation on chaperone dynamics, the authors should coIP Hsc82/Ssa2-4 in Fig 4E. If no specific antibodies for these isoforms exist, the authors could also express the different Ssa isoforms as the sole copy of Ssa in yeast (as they have done previously (Omkar, 2022)) and repeat Fig 4E in those cells.

Minor Comments:

Figure 1-3 are all directly related and therefore it would be more appropriate to combine them into a single figure if possible.

P11, line 237 should be Figure 5D, not 6D.

The authors should also mention the possibility of Lys PTMs aside from ubiquitination (e.g. SUMOylation) as potential modifiers of these conserved Ydj1 residues that would influence the chaperone interplay in higher eukaryotes.

Reviewer #2: The manuscript by Omkar S. et al. examines the role of lysine acetylation modification on the J-domain of Ydj1. The 12 acetylation sites were identified using the GPM database, and further work focused on the 6 sites present at the J- domain of the protein. The authors demonstrate that Ydj1 mutants that mimic non-acetylation support yeast growth like that of wild-type Ydj1 in response to a number of stress conditions. Among various mutants, the acetylation mimics of Lys23 and Lys37 poorly supported yeast growth against thermal stress. Further, the 6KQ mutant showed a loss of function phenotype against all stressors examined in the study. The authors further carried out an interactome study to examine the effect of acetylation on interaction with other cellular proteins, and found that acetylation affects Ydj1 interaction primarily with those involved in translation. The study further explored the role of Ydj1 acetylation in the stabilization of ribosomal subunits. Overall it's an important study that uncovers the role of Ydj1 acetylation on its native function and opens up various interesting possibilities to examine the role of PTMs on Ydj1 in the maintenance of protein homeostasis.

I have the following few comments before the manuscript is accepted for publication:

(1) The primary focus of the manuscript is to examine the significance of Ydj1 acetylation in its in vivo roles. However, no attempts are made to show that the co-chaperone is acetylated under any of the stress conditions examined. The authors should provide evidence of acetylation under at least one of the stress conditions examined.

(2) Figure 4E: Does this antibody bind specifically to Ssa1 or also to its other isoforms (Ssa2/3/4)?

(3) Figure 4E: The 6KQ mutant exhibits a weaker binding affinity to Ssa Hsp70 than the wild-type Ydj1. The binding of 6KR is much stronger. It would be uncommon for all six lys to undergo acetylation simultaneously in vivo. It is thus important to identify specific acetylation site(s) (among the 6 lysines) that affect Ydj1 interaction with Ssa.

(4) Figure 5C: The lowered abundance of Rps0A, Rps3 and Gcd11 could be due to an effect on transcription. Authors should explore this possibility.

(5) Page 11: “whereas the 6KQ and K37Q exhibited ~ 2 fold……….to wild type (Figure 6A)”. The mentioned mutants 6KQ and K37Q are most likely 6KR and K37R (needs to be clarified). Additionally, what could be the reason for the increased activity of these mutants in comparison to the wild-type protein, given that the wild-type protein is also not acetylated? 6KR has increased interaction with Ssa1 however K37R has similar interaction with Ssa1, and thus increased activity seems to be unrelated to changes in interaction with Ssa1.

(6) Page 12: Authors mention that stimulation of Ssa1 activity by 6KQ and wt Ydj1 is similar. However, Figure 6B shows that wt Ydj1 is significantly more active. This needs to be clarified.

Reviewer #3: see attachment

**Have all data underlying the figures and results presented in the manuscript been provided?**

Reviewer #1: Yes

Reviewer #2: Yes

Reviewer #3: **No: **Line 552: Data Availability. Files do not yet appear to have been deposited in MassIVE or PRIDE.

PLOS authors have the option to publish the peer review history of their article (what does this mean?). If published, this will include your full peer review and any attached files.

Reviewer #1: No

Reviewer #2: No

Reviewer #3: No

---

## [Decision Letter · Decision Letter 1]

4 Nov 2024

PGENETICS-D-24-00647R1Acetylation of the yeast Hsp40 chaperone protein Ydj1 fine-tunes proteostasis and translational fidelityPLOS Genetics Dear Dr. Truman, Thank you for submitting your manuscript to PLOS Genetics. After careful consideration, we feel that it has merit but does not fully meet PLOS Genetics's publication criteria as it currently stands. Therefore, we invite you to submit a revised version of the manuscript that addresses the points raised during the review process. Please submit your revised manuscript within 30 days Dec 04 2024 11:59PM. If you will need more time than this to complete your revisions, please reply to this message or contact the journal office at plosgenetics@plos.org. Please include the following items when submitting your revised manuscript:*
A rebuttal letter that responds to each point raised by the editor and reviewer(s). You should upload this letter as a separate file labeled 'Response to Reviewers'. This file does not need to include responses to formatting updates and technical items listed in the 'Journal Requirements' section below.*
A marked-up copy of your manuscript that highlights changes made to the original version. You should upload this as a separate file labeled 'Revised Manuscript with Track Changes'.*
An unmarked version of your revised paper without tracked changes. You should upload this as a separate file labeled 'Manuscript'. If you would like to make changes to your financial disclosure, competing interests statement, or data availability statement, please make these updates within the submission form at the time of resubmission. Guidelines for resubmitting your figure files are available below the reviewer comments at the end of this letter. We look forward to receiving your revised manuscript. Kind regards, Guang-Chao ChenAcademic EditorPLOS Genetics Fengwei YuSection EditorPLOS Genetics Aimée DudleyEditor-in-ChiefPLOS Genetics Anne GorielyEditor-in-ChiefPLOS Genetics **Journal Requirements:** **Additional Editor Comments (if provided):** While Reviewers 2 and 4 are satisfied with the revised manuscript, Reviewer 1 still has some concerns and comments that need to be addressed. Please consider and respond to these comments.**Reviewers' comments:** Reviewer's Responses to Questions

**Comments to the Authors:**

Reviewer #1: Regarding initial review Comment 1: Despite the framing of this paper around Ydj1 acetylation, the authors have not experimentally validated Ydj1 acetylation at any point in the manuscript. This is absolutely essential to support the interpretation of the results.

The authors must demonstrate Ydj1 acetylation in the presence and absence of the 6KR mutations. This is an essential piece of data that will determine whether the mutations have had any impact on the overall acetylation status of Ydj1. How can the authors be sure that mutation of these K residues has not induced compensatory acetylation at other K sites? This is especially salient given that only 11 K residues on Ydj1 are reported to be acetylated, and the authors mutate 6/11 in this study.

Regarding initial review Comment 3: The authors have consistently used the 6KR/6KQ mutants throughout the manuscript, however much of the change in activity is derived from mutation of K23 or K37. Perhaps a double mutant of these two residues would be more representative of Ydj1 acetylation state, and mitigate the potential effects of mutations at the other 4 sites. A more thorough exploration of different mutation combinations is necessary.

Mutating all 6 residues simultaneously could cause unforeseen changes in Ydj1 structure/function that could confound interpretation of the effect of the 2 most impactful mutations (especially in the context of the other 4 mutations). If acetylation at the other four sites is context-specific or of low stoichiometry, simultaneous mutations of all 6 residues is not an appropriate way to study the effects. This point is underscored by the observation that the mutant with the most prominent phenotype, K37, is also the most commonly observed acetylation site in Ydj1.

Reviewer #2: The queries have been satisfactorily answered. The manuscript is now recommended for publication.

Reviewer #4: The authors have addressed my comments with satidfactory changes and explanations. It is unfortunate that there are not suitable reagents available to easily detect the modifications without MS but presumably this will be reinforced by future studies.

**Have all data underlying the figures and results presented in the manuscript been provided?**

Reviewer #1: Yes

Reviewer #2: Yes

Reviewer #4: Yes

PLOS authors have the option to publish the peer review history of their article (what does this mean?). If published, this will include your full peer review and any attached files.

Reviewer #1: No

Reviewer #2: No

Reviewer #4: No

 **Figure resubmission:** While revising your submission, please upload your figure files to the Preflight Analysis and Conversion Engine (PACE) digital diagnostic tool, https://pacev2.apexcovantage.com/. PACE helps ensure that figures meet PLOS requirements. To use PACE, you must first register as a user. Registration is free. Then, login and navigate to the UPLOAD tab, where you will find detailed instructions on how to use the tool. If you encounter any issues or have any questions when using PACE, please email PLOS at figures@plos.org. Please note that Supporting Information files do not need this step. If there are other versions of figure files still present in your submission file inventory at resubmission, please replace them with the PACE-processed versions. **Reproducibility:** To enhance the reproducibility of your results, we recommend that authors deposit laboratory protocols in protocols.io, where a protocol can be assigned its own identifier (DOI) such that it can be cited independently in the future. Additionally, PLOS ONE offers an option to publish peer-reviewed clinical study protocols. Read more information on sharing protocols at https://plos.org/protocols?utm_medium=editorial-email&utm_source=authorletters&utm_campaign=protocols

---

## [Editor Report · Decision Letter 2]

25 Nov 2024

Dear Dr Truman,

We are pleased to inform you that your manuscript entitled "Acetylation of the yeast Hsp40 chaperone protein Ydj1 fine-tunes proteostasis and translational fidelity" has been editorially accepted for publication in PLOS Genetics. Congratulations!

Yours sincerely,

Guang-Chao Chen

Academic Editor

PLOS Genetics

Fengwei Yu

Section Editor

PLOS Genetics

Aimée Dudley

Editor-in-Chief

PLOS Genetics

Anne Goriely

Editor-in-Chief

PLOS Genetics

Comments from the reviewers (if applicable):

**Data Deposition**

http://datadryad.org/submit?journalID=pgenetics&manu=PGENETICS-D-24-00647R2

**Press Queries**

---

## [Editor Report · Acceptance letter]

28 Nov 2024

PGENETICS-D-24-00647R2 

Acetylation of the yeast Hsp40 chaperone protein Ydj1 fine-tunes proteostasis and translational fidelity 

Dear Dr Truman, 

We are pleased to inform you that your manuscript entitled "Acetylation of the yeast Hsp40 chaperone protein Ydj1 fine-tunes proteostasis and translational fidelity" has been formally accepted for publication in PLOS Genetics! Your manuscript is now with our production department and you will be notified of the publication date in due course.

With kind regards,

Anita Estes

PLOS Genetics

On behalf of:
